# CELO Fiber1 Knob Is a Promising Candidate to Modify the Tropism of Adenoviral Vectors

**DOI:** 10.3390/genes13122316

**Published:** 2022-12-08

**Authors:** Yangyang Sun, Xiaohui Zou, Xiaojuan Guo, Chunlei Yang, Tao Hung, Zhuozhuang Lu

**Affiliations:** 1School of Laboratory Medicine, Weifang Medical University, Weifang 261053, China; 2NHC Key Laboratory of Medical Virology and Viral Diseases, National Institute for Viral Disease Control and Prevention, Chinese Center for Disease Control and Prevention, Beijing 100052, China; 3Department of Experimental Technology, Henan Chemical Technician College, Kaifeng 475008, China

**Keywords:** fowl adenovirus 4, CELO virus, fiber, modification, gene transduction, vector, tropism

## Abstract

Fowl adenovirus 4 (FAdV-4) has the potential to be constructed as a gene transfer vector for human gene therapy or vaccine development to avoid the pre-existing immunity to human adenoviruses. To enhance the transduction of FAdV-4 to human cells, CELO fiber1 knob (CF1K) was chosen to replace the fiber2 knob in FAdV-4 to generate recombinant virus F2CF1K-CG. The original FAdV4-CG virus transduced 4% human 293 or 1% HEp-2 cells at the multiplicity of infection of 1000 viral particles per cell. In contrast, F2CF1K-CG could transduce 98% 293 or 60% HEp-2 cells under the same conditions. Prokaryotically expressed CF1K protein blocked 50% transduction of F2CF1K-CG to 293 cells at a concentration of 1.3 µg/mL while it only slightly inhibited the infection of human adenovirus 5 (HAdV-5), suggesting CF1K could bind to human cells in a manner different from HAdV-5 fiber. The incorporation of CF1K had no negative effect on the growth of FAdV-4 in the packaging cells. In addition, CF1K-pseudotyped HAdV-41 could transduce HEp-2 and A549 cells more efficiently. These data indicated that CF1K had the priority to be considered when there is a need to modify adenovirus tropism.

## 1. Introduction

Adenoviruses are a group of non-enveloped viruses, the virion of which packages a genome of linear double-stranded DNA of 26–48 kb in length. Adenoviridae consists of six genera, among which mastadenoviruses infect mammalian hosts exclusively while aviadenoviruses have been found only in birds [1]. Human adenoviruses (HAdV), especially HAdV-5, have been intensively studied. Adenoviruses have been isolated from humans, chimpanzees, monkeys, bovines, sheep, porcine, dogs, mice and other mammals. Some of them have been constructed as gene transfer vectors. Compared with other viral vectors, adenoviral vectors possess features such as stable and manipulable genome, scalable production and purification, efficient gene transfer and genetic safety [2,3]. Adenoviral vectors have been widely used in gene therapy and vaccine development [2,4].

Tropism is one of the most important properties needed to be considered when choosing an adenoviral vector [5,6]. Higher affinity to target cells means more efficient gene transfer, lower use dose and decreased side effect. Different adenoviruses have diverse cell tropism. However, the establishment of a novel adenoviral vector system is complicated, and a novel adenoviral vector does not always ensure an ideal cell targeting. That is why tropism modification is an essential aspect of adenovirus vector construction. There are some approaches to change the tropism of an adenovirus, including fiber knob substitution and incorporation of exogenous peptide to the fiber, pIX or hexon proteins [5,6]. Because fiber is the major ligand for adenovirus to recognize and bind its cellular receptor, fiber knob substitution is a practicable approach with an expectable outcome.

Pre-existing immunity is the major obstacle that hampers the application of adenoviruses in human gene therapy and vaccine inoculation [7,8]. Adenoviruses are ubiquitous pathogens in humans. For example, the prevalence of serum neutralizing antibody against HAdV-5 in adults is as high as 50–90%, which weakened the efficacy of HAdV-5 vector-based vaccines [9,10]. To overcome this obstacle, efforts have been taken to construct vectors based on rare serotypes, non-human adenoviruses or even fowl adenoviruses [11,12,13,14,15]. In the 1970s, serological studies have found and confirmed that aviadenoviruses did not share group antigens with mastadenovirusies, providing the foundation to develop fowl adenoviral vectors to evade the pre-existing immunity to HAdV in human beings [16].

In previous studies, we established a vector system based on fowl adenovirus 4 (FAdV-4) and evaluated the effect of genus-specific gene deletions on virus growth [17,18,19,20]. The gene transfer ability of FAdV-4 vector to human cells was very low. We tried to insert RGD4C peptide to the fiber1 knob and replace the knob of fiber2 with that from HAdV-35 fiber, such modification substantially improved the gene transduction of FAdV-4 to human adherent or suspension cell lines [21,22,23]. However, the activity of current FAdV-4 vectors still did not meet the requirement of transducing human cells.

Chick embryo lethal orphan virus (CELO) was the prototype of fowl adenovirus A1 [14]. It was reported that CELO fiber1 could bind the CAR molecule on human cells. However, evidence from fiber structure studies did not support such conclusion [24,25]. CELO has been constructed as a gene transfer vector and it had the capability to transduce human cells [14]. Here, we attempted to replace the knob of FAdV-4 fiber2 with that of CELO fiber1 and evaluate the possibility of using the modified FAdV-4 as a vector to transduce human cells. In addition, we further assessed the function of CELO fiber1 knob (CF1K) in HAdV-41 and proposed that CF1K could be a good option when there was a need to modify adenovirus tropism.

## 2. Materials and Methods

### 2.1. Cells, Viruses, Plasmids and Oligonucleotides

Flasks or plates were pre-treated with 0.1% gelatin (Cat. G9391, Sigma-Aldrich, St. Louis, MO, USA) in water at 37 °C for half an hour before seeding LMH cells according to the instruction of American Type Culture Collection (ATCC). Chicken LMH cells (Leghorn Male Hepatoma; ATCC CRL-2117), human 293 (ATCC CRL-1573), HEp-2 (ATCC CCL-23), A549 (CCL-185) and 293TE32 cells were cultivated in Dulbecco’s modified Eagle’s medium (DMEM) containing 10% fetal bovine serum (FBS; HyClone, Logan, UT, USA) at 37 °C in a humidified atmosphere supplemented with 5% CO_2_ and routinely split twice a week. DMEM containing 2% FBS was used for virus rescue or amplification. 293TE32 cells are a derivative strain of 293 cells, which constitutively expressed HAdV-41 E1B55K protein and served as the packaging cells for replication-defective HAdV-41 viruses [26,27]. Exponentially growing cells were split 1:3 and would be ready for plasmid transfection or virus infection when they reached 70–90% confluency the next day. FAdV4-CG and HAdV41-CG were recombinant adenoviruses constructed previously in the laboratory. FAdV4-CG is a fowl adenovirus 4 (FAdV4) carrying deletions of ORF1, ORF1b and ORF2 and an insertion of CMV promoter controlled GFP (CG) expression cassette at the left side of the genome [18]. HAdV41-CG originated from human adenovirus 41, and the E1 and E3 regions in the genome were deleted and replaced with CMV promoter (CMVp) controlled GFP expression cassette and HAdV-5 ADP gene, respectively [26]. pKFAV4S-GFP is an adenovirus plasmid, and it carries deletions of ORF0, ORF1, ORF1B, ORF2 and ORF19A in FAdV-4 genome; and at the left deletion site in the genome, a CMVp controlled GFP expression cassette was inserted, which was flanked with two SwaI sites to facilitate transgene replacement [20]. pMD-FAV4Fs is an intermediate plasmid for FAdV-4 fiber modification, which contains fiber1 and fiber2 genes of FAdV-4 [19]. Single-stranded DNA oligos were chemically synthesized and used in PCR. The information related to PCR templates and primers was summarized in Table 1.

### 2.2. Construction of Adenoviral Plasmids

An intermediate plasmid-based strategy was employed for FAdV-4 fiber substitution, and we called the cloning method of combined restriction digestion and Gibson assembly restriction-assembly [28,29]. Briefly, two unique cutter (AvrII/HindIII) sites were chosen after restriction analysis of the intermediate plasmid pMD-FAV4Fs in silico, which flanked fiber2 knob region. PCR was performed to amplify four fragments: F2-AvrII covered the region from AvrII site to the knob-knob substitution point in pMD-FAV4Fs, CF1K and T-KpnI covered the CELO fiber1 knob (the KpnI site on the conjunction was synonymously mutated), and the F2-HindIII covered the region from the knob-knob substitution point to HindIII site in pMD-FAV4Fs. These four fragments were ligated together to generate the fragment containing CF1K by overlap extension PCR. CF1K-containing fragment was inserted into the AvrII-HindIII sites in pMD-FAV4Fs to generate pMD-FAV4S-F2CF1K by restriction-assembly (Appendix A). The modified fiber region in pMD-FAV4S-F2CF1K was restored to pKFAV4S-GFP to generate adenoviral plasmid pKFAV4S-F2CF1K-CG. The EF1a promoter and GFP coding sequence (EG-SwaI) were amplified and inserted into the two SwaI sites to replace CMVp-GFP in pKFAV4S-F2CF1K-CG to generate pKFAV4S-F2CF1K-EG by restriction-assembly [30]. The adenoviral plasmid of pKAd41-CF1K-CG was similarly constructed (Appendix A and Table 1).

### 2.3. Virus Rescue, Purification, Titration and Identification

pKFAV4S-F2CF1K-CG was linearized by PmeI digestion and used to transfect LMH cells. Plaques were observed under fluorescence microscope 3 to 5 days post-transfection. The rescued viruses were further amplified in LMH cells. When complete cytopathic effect (CPE) occurred, the culture media and cells were separately harvested. Viruses in the cells were released by three rounds of freeze-and-thaw, separated from the cellular debris by centrifugating at 2500× *g* for 10 min, treated with Benzonase Nuclease (New England Biolabs, Ipswich, MA, USA) at a final concentration of 50 U/mL at room temperature for 1 h and centrifugated at 1500× *g* for 10 min to remove debris. Citrate buffer (0.5 M, pH 6.0) was added to the culture media to a final concentration of 10 mM, and ammonium sulphate powder was added to reach 40% saturation. The culture media was let stand at room temperature for 8–12 h before centrifugating at 2000× *g* for 15 min [31]. The supernatant was discarded and the precipitate was dissolved in 10 mM citrate buffer. Iodixanol gradients (Opti-prep, Axis-Shield PoC AS, Oslo, Norway) of 50%, 40%, 35%, 30% and 25% were prepared in a buffer containing 10 mM citrate, 2.5 mM KCl, 0.04% Tween-80 and 150 mM NaCl (pH 6.0), and a 15% gradient was similarly prepared except that it contained 1.15 M NaCl [32,33]. These gradients were loaded to tubes sequentially, and the virus solutions derived from cells or culture media were finally loaded on the top. The loaded tubes were centrifugated at 100,000× *g* for 2 h, virus bands (between the 40% and 50% gradient layers) were collected, 60% glycerol was added to reach a final concentration of 5%, and the virus suspension was thoroughly mixed, aliquoted and stored at −80 °C [17]. Recombinant HAdV-41 viruses were rescued and amplified in 293TE32 cells, purified according to the protocol of traditional CsCl gradient ultracentrifugation [34], and stored in a buffer containing 1 mM MgCl_2_, 150 mM NaCl, 10 mM Tris-Cl and 5% glycerol (pH 7.6).

The virus particle titer was determined by measuring the content of virus genomic DNA. Briefly, purified virus was mixed with an equal volume of lysis buffer containing 20 mM EDTA, 1% SDS and 0.4 mg/mL proteinase K (pH 7.6) and incubated in a water bath at 50 °C for 2 h. The lysate was then diluted with water, and the DNA content was determined with the Qubit double-stranded DNA assay kit (Cat. Q32851, Thermo Fisher Scientific, Waltham, MA, USA) [35], and the result was further confirmed by resolving the virus DNA on an agarose gel with a quantified lambda DNA/Hind III digest as the control [36]. The mass concentration was converted to molar concentration after being divided by the molecular weight of adenoviral genome, and the viral particle titer (the concentration of viral genome copies) was calculated by multiplying the molar concentration with Avogadros constant. The multiplicity of infection (MOI) was calculated from particle titers. Infectious titers (IU/mL) were determined with the limiting dilution assay on LMH or 293 cells by counting GFP+ cells one or two days after infection [36]. The virus was identified by restriction analysis: virus genomic DNA was extracted from lysed purified virus (Genomic DNA Clean & Concentrator kit, Cat. D4010; Zymo Research, Irvine, CA, USA), digested with restriction enzymes, resolved on 0.7% agarose gel containing ethidium bromide by electrophoresis and visualized on a UV transilluminator. The modification region was amplified by PCR and confirmed by sequencing.

### 2.4. Gene Transduction Assay

Viruses were diluted in DMEM containing 2% FBS and used to infect cells in 24-well plate at a volume of 0.25 mL/well. The plate was placed on a mixer at 37 °C and rocked for 4 h at a frequency of 10 times per minute. The virus diluent was removed and 0.5 mL fresh DMEM plus 2% FBS was added to each well. Forty-eight hours post-infection (calculated from the start of virus incubation), GFP expression was observed and photographed under fluorescence microscope, and the cells were detached by trypsin treatment, dispersed into single cells, suspended in 10 mM phosphate buffered saline (PBS) containing 1% FBS and 1.5% paraformaldehyde and reserved at 4 °C temporarily. The fixed cells were subjected to flow cytometry assay within one week.

### 2.5. Infection Blocking Experiment

CF1K, together with the coding sequence for knob-shaft linker, was amplified by PCR and inserted into the NdeI/HindIII sites in pET-30a(+) prokaryotic expression vector to generate pET30-HCF1K plasmid by restriction-ligation cloning. The fusion of 6×His tag with CF1K (HCF1K) was confirmed by sequencing before pET30-HCF1K was transformed into E. coli BL-21(DE3) chemically competent cells. The detailed procedure of HCF1K expression and purification can be found elsewhere [19]. The purified protein was dialyzed against 10 mM PBS to remove imidazole, aliquoted and quantified with BCA protein assay (Cat. KGPBCA, KeyGEN Biotech, Jiangsu, China). The protein was diluted in water, mixed with 2× SDS gel-loading buffer (100 mM Tris-Cl, 4% sodium dodecyl sulphate, 0.2% bromophenol blue, 20% glycerol, pH 6.8) with or without 200 mM dithiothreitol. The dithiothreitol-included sample was heated at 95 °C for 10 min, and the other was unheated. Together with control samples, they were loaded and resolved on precast SDS-PAGE gel (4–20%, Cat. P0057A, Beyotime Biotechnology, Beijing, China) before Coomassie Brilliant Blue R-250 Staining.

HCF1K protein of 180 µg was added to 5 mL DMEM plus 2% FBS, sterilized by passing through a 0.22 µm membrane filter, and serially diluted with DMEM plus 2% FBS. Human embryonic kidney 293 cells were seeded in 24-well plate the day before, old media were removed, diluted HCF1K was added to each well at a volume of 0.25 mL and the recombinant protein was let adsorb to the cells by rocking the plate at room temperature for 30 min. After that, 50 µL diluted FAdV4-F2CF1K-CG (F2CF1K-CG) viruses in DMEM plus 2% FBS was added to each well to reach an MOI of 200 vp/cell. HAdV5-CG was similarly applied to other wells to reach an MOI of 20 vp/cell and served as the control. After the plates were placed at the room temperature without disturbance for 60 min, the HCF1K and virus mixture was discarded, the cells were washed twice with PBS, 0.5 mL DMEM plus 5% FBS was added to each well, and the plates were transferred to the incubator and cultivated at 37 °C. Then, 24 h later, the culture media were aspirated, and the cells were washed once with PBS, fixed with 4% paraformaldehyde in PBS (200 µL per well), and photographed under fluorescence microscope. GFP fluorescence intensity was further determined on a multimode plate reader (Infinite M1000 PRO, Tecan Austria GmbH, Grödig, Austria). The Excitation and Emission wavelength were set to be 488 nm and 509 nm, respectively, and multiple reads of 7 × 7 per well were applied. The average readout from wells with uninfected cells was treated as background and subtracted from the readout of test wells.

### 2.6. Plaque Forming Assay

Adenoviruses of 5000 or 10,000 vp were diluted in 1 mL DMEM plus 2% FBS and used to infect LMH cells seeded in 6-well plate for 2 h. The virus diluents were discarded, the cells were washed twice with DMEM, and 2.5 mL semisolid DMEM media containing 2% FBS and 1% low-melting agarose were added to each well. Two mL fresh DMEM containing 2% FBS was supplemented to provide more nutrients 4 days post-infection. The GFP foci were photographed under a fluorescence microscope with a 4× objective lens mounted at 5 days post-infection. The area of all foci in half well (for F2CF1K-CG) or two replicate wells (for FAdV4-CG) in the culture plate was measured by using the Fiji image processing package (http://fiji.sc/ accessed on 1 March 2021) [37]. The median sizes of the foci formed by different viruses were compared by using the Mann-Whitney nonparametric test. At day 6 post-infection, liquid culture medium was removed carefully, 2.5 mL 4% paraformaldehyde in PBS was added to the top of semi-solid medium. Cells were fixed for 4 h at room temperature and subsequently stained with crystal violet solution [38].

### 2.7. One-Step Growth Curve

LMH cells in 12-well plates were infected with F2CF1K-CG or FAdV4-CG at an MOI of 5 vp/cell in 0.5 mL DMEM plus 2% FBS for 2 h. After removal of the virus diluent, the cells were washed twice with DMEM and cultured in 1 mL DMEM plus 2% FBS at 37 °C. At indicated time points, culture medium of 600 µL were transferred to a 1.5 mL tube, centrifugated at 300× *g* for 5 min, the supernatant of 400 µL was transferred to a new tube, the cells in the well were detached by scraping and aspiration, the detached cells and culture medium were transferred to the tube that contained the remnant 200 µL medium, and all the tubes were temporarily stored at −80 °C. The tubes containing cells were subjected into three rounds of freeze-and-thaw and centrifugated at 5000× *g* for 5 min, and the supernatant was titrated. The clarified medium (0.4 mL) was thawed and titrated. The number of viruses released to the medium was calculated by multiplying the titer value of the clarified medium with the volume of 1 mL. The titer of clarified medium was subtracted from that of the cell-medium mixture, and the answer times the volume of 0.6 mL was the number of cell-associated viruses.

### 2.8. Detection of Virus Genome in 293 Cells

In total, 293 cells in 6-well plates were infected with F2CF1K-CG or HAdV5-CG at MOIs of 20, 100 or 500 vp/cell at 37 °C for 2 h. The viruses were removed, and cells were washed and cultivated in 2 mL DMEM plus 2% FBS. At indicated time points, the media were discarded, and the cells were washed twice with PBS and scraped off with a cell lifter. The genomic DNA from cells and viruses were extracted (Cat. no. DP307, TIANamp genomic DNA kit, TIANGEN, Beijing, China). Taqman probe-based real-time PCR was performed to determine the copy number of viral genome with the primers and probe targeting GFP gene (Table 1). The cell number was calculated by determining the copy number of cellular RNase P gene by real-time PCR [39].

## 3. Results

### 3.1. Construction of CELO Fiber1 Knob-Pseudotyped Adenoviruses

An intermediate plasmid-based strategy was employed to modify the fiber gene in adenoviral genome [28,29]. In brief, a fragment, which contained the region to be modified, was excised from adenoviral genome to form a smaller intermediate plasmid. Because more unique restriction enzyme sites can be found and used in the intermediate plasmid, the target gene will be conveniently modified with overlap extension PCR. The modified fragment is restored to generate a new adenoviral plasmid by restriction-assembly. The procedure of constructing intermediate plasmid pMD-FAV4FS-F2CF1K was shown in Appendix A, and the schematic diagram of constructing CF1K-pseudotyped FAdV-4 was shown in Figure 1A. The conjunction site between the shaft domain of FAdV-4 fiber2 and the knob region of CELO fiber1 was shown in detail in Figure 1B. The generated pKFAV4S-F2CF1K-CG and the parental plasmid pKFAV4S-GFP were designed to be a single plasmid-based adenoviral vector system, and the transgene expression cassette was flanked with dual cutter (SwaI) sites and could be conveniently replaced with other promoters or transgenes [30]. Therefore, the EF1a promoter-controlled GFP fragment was amplified by PCR and inserted into the two SwaI sites to generate pKFAV4S-F2CF1K-EG by restriction-assembly.

Adenoviral plasmids were linearized and transfected to chicken LMH cells. The GFP foci were found 2 days post-transfection. They kept growing and formed plaques in the following days (Figure 2A). The viruses were amplified and purified by density gradient ultracentrifugation (Figure 2B). The genomic DNA was extracted and identified by restriction analysis (Figure 2C,D). The modified regions were further confirmed by sequencing. Short names of F2CF1K-CG and F2CF1K-EG were used to represent these two modified FAdV-4 viruses in following description.

The knob of HAdV-41 short fiber was similarly replaced with CF1K to generate recombinant HAdV41-CF1K-CG virus (Appendix A). The titer and yield of these viruses were summarized in Table 2.

The genetic background of FAdV-4 vectors was described in the Materials and Methods section, while HAdV-41 vectors carried deletion in E1/E3 region in the viral genome. All vectors carried GFP reporter gene. F1IJR-CG: FAdV4-F1IJR-CG; F2CF1K-CG: FAdV4-F2CF1K-CG; F2CF1K-EG: FAdV4-F2CF1K-EG; RGD4C peptide: CDCRGDCFC [22]; vp: viral particle; IU: infectious unit.

### 3.2. CELO Fiber1 Knob-Pseudotyped FAdV-4 Transduced Human Adherent Cell Lines Efficiently

Original FAdV-4 vectors could hardly transduce human cells. When FAdV4-CG, a recombinant FAdV-4 carrying GFP reporter gene, was used at an MOI of 1000 vp/cell, less than 4% of 293 cells were detected to express GFP [21]. In order to increase the transduction of FAdV-4 to human cells, RGD4C peptide was inserted to the CD, DE, HI and IJ loops of FAdV-4 fiber1 knob to generate fiber-modified viruses in a previous study [21]. FAdV4-F1IJR-CG (F1IJR-CG) was superior to others and it could transduce about 70% 293 cells when being used at an MOI of 1000 vp/cell. Therefore, F1IJR-CG served as a control in this study.

The transduction of fiber-modified FAdV-4 vectors to human 293, HEp-2 and A549 cells was investigated (Figure 3). At an MOI of 100 vp/cell, F1IJR-CG transduced 16% 293 cells while F2CF1K-CG transduced 60%. When the MOI increased to 1000 vp/cell, the efficiencies of transducing 293 cells were 75% and 98% for F1IJR-CG and F2CF1K-CG, respectively. If we assume that the transduction efficiency is proportional to MOI value when the transduction efficiency is lower than 75%, it was inferred that the transduction rate could be 60% when F1IJR-CG was applied at an MOI of 770 vp/cell. To reach a transduction rate of 60%, an MOI of 100 vp/cell was needed for F2CF1K-CG. Therefore, it might be deduced that F2CF1K-CG was about seven times more efficient than F1IJR-CG in transducing 293 cells. Similarly, for HEp-2 cells, F2CF1K-CG or F2CF1K-EG were about four or ten times more efficient than F1IJR-CG, respectively. However, F2CF1K vectors were not superior to F1IJR-CG in A549 cells (Figure 3B). Notably, the transduction activity of F2CF1K-CG was still inferior to that of HAdV-5 vector. For HEp-2 cells, HAdV5-CG was about seven times more efficient than F2CF1K-CG. All FAdV-4 vectors could efficiently infect LMH cells.

### 3.3. F2CF1K-CG Bound to 293 Cells in a Manner Different from HAdV5-CG

HCF1K (6×His tagged CF1K) was prokaryotically expressed. The soluble form was purified with the method of immobilized metal ion affinity chromatography (IMAC). HCF1K was a trimer [24], which meant each trimeric molecule contained three 6×His tags and could bind chromatography media with high affinity. Therefore, high concentration of imidazole was included in the balance and washing buffer, and impurities were successfully removed (Figure 4A).

Human embryonic kidney 293 cells were incubated with HCF1K at serial concentrations and infected with F2CF1K-CG or HAdV5-CG, since 293 was the most sensitive human cell line to the transduction of fiber-pseudotyped FAdV-4 (Figure 3). Gene transduction was recorded at 24 h post-infection. At this time point, HAdV5-CG did not complete its replication cycle and would not spread to neighboring cells. For F2CF1K-CG, the inhibition effect was detected at the concentration of 49 ng/mL, and it became more remarkable as more HCF1K was added. When the concentration increased to 1.3 µg/mL, more than half of GFP expression was inhibited. HCF1K could interrupt the infection of HAdV5-CG when being used at a higher concentration. The 50% inhibition concentration was about 36 µg/mL for HAdV5-CG, which was approximately 30 times higher than that for F2CF1K-CG (Figure 4B,C). Similar results were obtained from human HEp-2 cells, where 1.3 µg/mL HCF1K prevented 60% cells from the infection of F2CF1K-CG while the concentration needed to be increased to 12 µg/mL to achieve the same inhibition rate for HAdV5-CG (Appendix A). The cellular receptor of HAdV-5 was CAR (coxsackie virus and adenovirus receptor). These results implied that HCF1K could not bind CAR with high affinity to block the infection of HAdV-5, or HCF1K bound to CAR at a site different from that used by HAdV-5 fiber.

### 3.4. The Viral Genome Did Not Replicate in FAdV-4 Virus Infected 293 Cells

Human 293 cells were HAdV-5 E1-transformed embryonic kidney cells [40]. The 293 cells constitutively expressed E1 genes of HAdV-5 and could sustain the replication of many replication-competent or -defective adenoviruses. We investigated if the viral genome could replicate in 293 F2CF1K-CG-infected cells. The cells were infected with F2CF1K-CG at MOIs of 100 or 500 vp/cell for 2 h. At such doses, viral genomes entered more than 40% or 80% of 293 cells. Replication of viral genome was not detected. In fact, as the culture time prolonged, the copy number of F2CF1K-CG in cells decreased due to degradation. In contrast, the copy number of HAdV5-CG genome increased four orders of magnitude after two days’ culture, although the start amount of the viral genome was relatively lower (Figure 5). These data demonstrated that FAdV-4 genome was not able to replicate in 293 cells.

### 3.5. The Amplification of F2CF1K-CG in Chicken LMH Cells

The production of a vector needed to be considered for its application in gene therapy or vaccine development. The replication of F2CF1K-CG in LMH cells was evaluated by plaque-forming experiment and one-step growth curve. F2CF1K-CG formed smaller plaques than FAdV4-CG did, but the number of formed plaques was much higher on F2CF1K-CG-infected LMH cells (Figure 6A,B). When LMH cells were infected with F2CF1K-CG or FAdV4-CG at a low MOI of 5 vp/cell, the progeny viruses were constantly produced in the following 5 days. The amplification of F2CF1K-CG was much faster and reached its peak at 60 h post-infection. It took about 96 h for FAdV4-CG to achieve the yield peak. It looked like that F2CF1K-CG produced more progenies since higher yield of infectious units were observed on the F2CF1K-CG curve. If the ratios of particle to infectious unit were taken into consideration (the ratio for FAdV4-CG was about ten times as high as that for F2CF1K-CG), the virion yields of these two viruses were very close (Table 2 and Figure 6C). Another conclusion could be drawn from these curves: cell lysis occurred and most of the progeny viruses were released to the culture media during the late phase of the infection.

### 3.6. Modification of HAdV-41 Short Fiber with CELO Fiber1 Knob

To observe the synergistic effect of CF1K and a CAR-binding fiber, the knob of HAdV-41 short fiber was replaced with CF1K. The rescued virus HAdV41-CF1K-CG grew efficiently in HAdV-41 packaging cells, and the virus was purified. HAdV41-CF1K-CG was slightly superior to the control HAdV41-CG when being used at a low MOI of 100 vp/cell on 293 cells. The difference between these two viruses became more obvious and significant when testing on HEp-2 or A549 cells (Figure 7A,B). By taking the assumption that percentage of GFP+ cells was proportional to the MOI value when the transduction rate was low, it could be calculated that the needed amount of HAdV41-CF1K-CG was one-third that of HAdV41-CG to achieve an identical transduction efficiency on HEp-2 cells. For A549 cells, an amount four times less of HAdV41-CF1K-CG was needed to reach the same transduction rate. These data indicated that incorporation of CF1K increased the ability of HAdV-41 to transduce HEp-2 and A549 cells.

## 4. Discussion

FAdV-4 has the potential to be constructed as a useful vector system, and FAdV-4 vector could have some advantages over its mammalian analogues. First, the cloning capacity of FAdV-4 could be larger. The genome of FAdV is about 8–10 kb longer than that of HAdV. The regions encoding genus-specific genes are approximately 7 kb on the left and 10 kb on the right in FAdV-4 genome, and no genus-specific gene is essential for the virus replication [20]. In F2CF1K-CG or FAdV4-CG, about 4.7 kb of virus genome has already been deleted. By expressing some important viral genes in packaging cells, it can be expected that exogenous DNA longer than 10 kb could be loaded to FAdV-4 genome. Second, the yield of FAdV-4 vectors was very high. If the progeny viruses released to the culture media were also harvested, LMH cells cultured in five 15 cm dishes produced progeny FAdV-4 viruses with a yield of more than 10^12^ vp. Third, FAdV-4 vectors could be safer for human gene therapy or vaccine development since they did not replicate in human cells. As being examined in this report, the viral genome did not replicate in human 293 cells (Figure 5). Finally, the use of FAdV-4 vector could avoid the disadvantage of pre-existing immunity against human adenoviruses. However, these advantages could be possibly realized only in conditions that FAdV-4 was capable of transducing human cells.

Fiber modification provided an approach to change the tropism of adenovirus. In previous studies, RGD4C peptide was inserted to several sites of FAdV-4 fiber1 knob, and the knob of HAdV-35 fiber was used to replace that of FAdV-4 fiber2. These modifications significantly improved the transduction of FAdV-4 to human cells [21]. However, high dose of vectors was still needed to achieve considerable gene transfer efficiency. The knob of FAdV-4 fiber2 was replaced with CF1K in this report, the generated F2CF1K-CG vector could efficiently transduce 293 or HEp-2 cells. When being used at the dose of 1000 vp/cell, the original FAdV4-CG transduced 4% 293 or 1% HEp-2 cells while the modified F2CF1K-CG could transduce 98% 293 or 63% HEp-2 cells. In fact, fowl adenovirus F2CF1K-CG was already superior to human HAdV41-CG in transducing these two cell lines (Figure 3 and Figure 7). The construction of F2CF1K-CG provided a solid foundation for further studying the behavior of fowl adenovirus in human cells.

CF1K has the priority to be considered when there is a need to modify adenovirus tropism for the following reasons: first, CF1K could significantly improve the gene transduction to adherent human cells, which was well exemplified in the construction and use of F2CF1K-CG. Second, CF1K can complement the function of the fiber of HAdV-5 or other adenoviruses. Many human adenoviruses including HAdV-5 used CAR as the cellular receptor [41,42]. At the beginning, the receptor of CF1K on human cells was also reported to be CAR [43]. However, crystal structure analysis pointed out there was little chance for CF1K to bind CAR in a manner as HAdV fibers did [24,25]. Purified CF1K protein inhibited the transduction of F2CF1K-CG virus, but it had minimal effect on the infection of HAdV-5 to 293 cells (Figure 4), supporting the conclusion from crystal structure analysis. There were two possibilities to explain the observed experimental phenomena: CF1K and HAdV-5 fiber bound different receptors on human cells or different sites on the extracellular domain of CAR molecule. In both cases, CF1K provided an attachment to human cells different from the fiber of HAdV-5 or other HAdVs. The cellular receptor for HAdV-41 long fiber is believed to be CAR while the receptor of the short fiber remains unknown [41,42]. In addition, the short fiber contributes little to the adsorption of HAdV-41 to cells [44]. Because HAdV-41 lacks the RGD motif in the penton base, HAdV-41 transduces human cell lines in vitro with a low efficiency [45]. Therefore, HAdV-41 is an ideal model to test the complemental effect between CF1K and a CAR-binding fiber. After replacing the knob of short fiber with CF1K, the transduction was improved (Figure 7), indicating a synergistic effect of CF1K to HAdV-41 long fiber. Finally, phylogenetic analysis indicated that the genetic distance between CF1K and fiber knob of human adenoviruses was much longer than that between human adenoviruses, implying that use of CF1K might help the vector partly evade the neutralization of pre-existing immunity [25]. Hexon and fiber are the most important neutralizing antigens that elicit humoral immune response [46,47]. It was reported that hexon elicited type-specific neutralizing antibodies whereas fiber induces more cross-neutralization between types [46]. There might be a greater payoff to choose a fiber knob distant in evolution.

The incorporation of CF1K had no negative effect on the growth of adenoviruses in packaging cells. F2CF1K-CG viruses formed smaller plaques on LMH cells, which might result from the enhanced affinity of the virus to host cell. In the environment of semisolid media, the majority of the released progeny viruses bound to the neighboring cells firmly and quickly and has little chance to spread to remote places. On the other hand, higher affinity caused more host cells being infected in a relatively short incubation and resulted in more plaques (Figure 6A,B). In a liquid culture environment, F2CF1K-CG could even destroy the monolayer of host cells in a shorter period and the total yield of progeny viruses was higher than that of the virus with native fibers (Figure 6C). For the HAdV-41 carrying CF1K, the situation was similar (Table 2).

The behavior of FAdV in human cells deserves further study. FAdV provides an ideal counterpart to human adenovirus, and this work on comparative adenovirology will provide opportunities to answer many interesting questions associated with adenoviruses, such as virus entrance, viral genome transport in plasma, background expression of viral genome, viral genome replication and the host-specific viral replication [48]. These studies will benefit the basic virology of adenovirus and gene therapy. F2CF1K-CG could be the appropriate model virus thanks to its high transduction to human cells.

## Figures and Tables

**Figure 1 genes-13-02316-f001:**
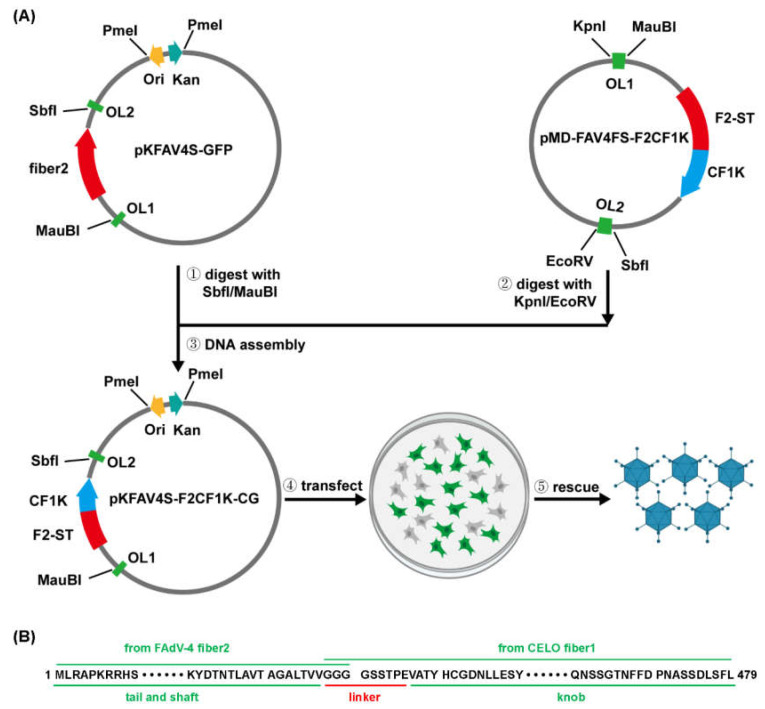
Schematic diagram of fiber modification procedure and sites for fowl adenovirus 4 (FAdV-4). (**A**) Procedure for substitution of FAdV-4 fiber2 knob. The intermediate plasmid pMD-FAV4Fs and adenoviral plasmid pKFAV4S-GFP were constructed previously, and their background was explained in detail in the Materials and Methods section. The fiber2 knob in pMD-FAV4Fs was replaced with fiber1 knob of CELO virus to generate intermediate plasmid pMD-FAV4FS-F2CF1K by site-directed mutation with overlap extension PCR. (**B**) Amino acid sequence of the fused fiber protein. CF1K: CELO fiber1 knob; F2-ST: FAdV-4 fiber2 shaft and tail; OL1: overlap1; OL2: overlap2.

**Figure 2 genes-13-02316-f002:**
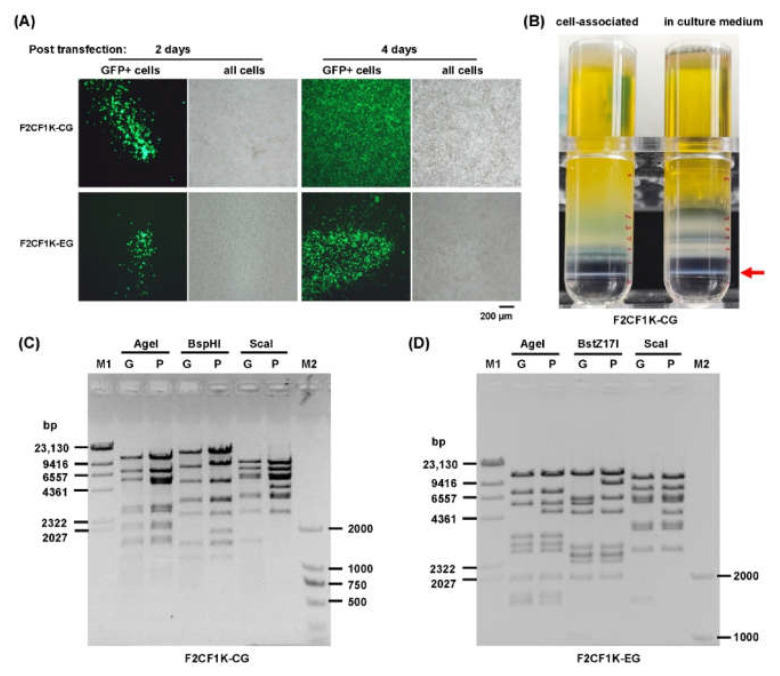
Rescue, purification and identification of fiber-modified FAdV-4 vectors. (**A**) Rescue of FAdV4-F2CF1K-CG and FAdV4-F2CF1K-EG virus. PmeI-linearized pKFAV4S-F2CF1K-CG or pKFAV4S-F2CF1K-EG were used to transfected LMH cells. Foci formed by GFP-positive cells were observed under a fluorescence microscope 2- or 4-days post-transfection, suggesting successful virus rescue. F2CF1K-CG and F2CF1K-EG are short names for these two recombinant viruses. (**B**) Purification of fiber-modified FAdV-4 by density gradient ultracentrifugation. Viruses were amplified in LMH cells, and the progeny viruses released to the media were first condensed by ammonium sulphate precipitation. Raw virus suspensions collected from infected cells or culture media were loaded on iodixanol gradients and centrifugated at 100,000× *g* at 8 °C for 2 h. The result of F2CF1K-CG was shown. The band of complete virions were indicated with an arrow. (**C**,**D**) Identification of adenoviral genome by restriction analysis. Virus genomic DNA was digested with restriction enzymes and resolved on 0.7% agarose gel by electrophoresis, and the corresponding adenoviral plasmid served as the control. The predicted molecular weights (bp) of digested fragments of F2CF1K-CG genome were 11,688, 7081, 5495, 2854, 2623, 2077, 1946, 1598, 1507, 1476, 1082 for AgeI; 14,730, 8092, 5113, 3296, 3185, 2474, 1440, 1395 for BspHI; and 9473, 7406, 5942, 5522, 3641, 3398, 2561, 1461 for ScaI. The predicted molecular weights (bp) of digested fragments of pKFAV4S-F2CF1K-CG plasmid were 11,688, 7081, 5495, 5060, 2854, 2623, 2077, 1946, 1598, 1476 for AgeI; 14,730, 8892, 5113, 3296, 3185, 2474, 1786, 1440, 1395 for BspHI; and 9473, 7406, 5942, 5522, 4377, 3641, 3398, 2561 for ScaI (**C**). The predicted molecular weights (bp) of digested fragments of F2CF1K-EG genome were 11,688, 7081, 5495, 3168, 2854, 2623, 1946, 1598, 1507, 1476 for AgeI; 12,141, 6118, 5557, 4639, 2755, 2545, 2494, 2337, 1956, 1018 for BstZ17I; and 10,194, 7406, 5942, 5522, 3641, 3398, 2561, 1461 for ScaI. The predicted molecular weights (bp) of digested fragments of pKFAV4S-F2CF1K-EG plasmid were 11,688, 7081, 5495, 4690, 3168, 2854, 2623, 1946, 1598, 1476 for AgeI; 12,141, 9046, 6118, 4639, 2755, 2545, 2494, 2337,1956 for BstZ17I; and 10,194, 7406, 5942, 5522, 4377, 3641, 3398, 2561 for ScaI (**D**). Molecular weights of fragments less than 1000 bp were not given.

**Figure 3 genes-13-02316-f003:**
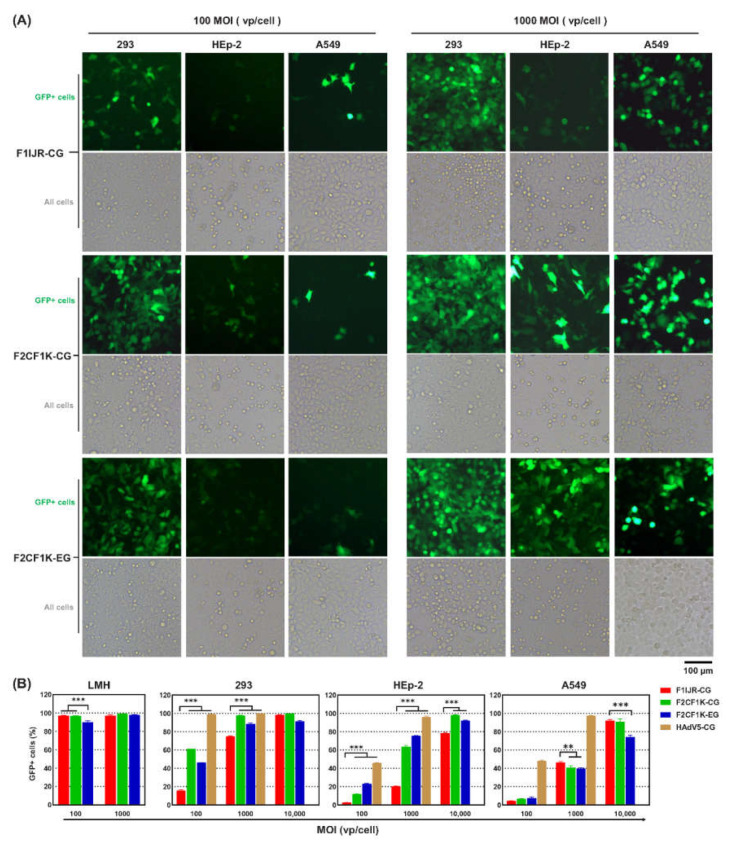
Transduction of human adherent cell lines with fiber-modified FAdV-4 vectors carrying GFP reporter gene. (**A**) Human 293, HEp-2 and A549 cells were infected with FAdV4-F2CF1K-CG (F2CF1K-CG) or FAdV4-F2CF1K-EG (F2CF1K-EG) viruses at various MOIs for 4 h. GFP expression was observed under a fluorescence microscope at 48 h post-infection. FAdV4-F1IJR-CG (F1IJR-CG), a previously constructed FAdV-4 carrying RGD4C-modified fiber1, served as a control. F1IJR-CG had an improved capability to transduce human cells when compared with original FAdV-4 vector [21]. (**B**) Quantitative analysis of GFP+ cells by flow cytometry. Cells were detached by trypsin treatment, dispersed into single cells and fixed in 1.5% paraformaldehyde at 2 days post-virus infection, and the expression of GFP was determined by flow cytometry. In some tests, recombinant HAdV-5 carrying CMV promoter controlled GFP expression cassette (HAdV5-CG) was used as the positive control. All of the experiments were performed in duplicate and the data shown are from one representative experiment out of the three performed. ** *p* < 0.01, *** *p* < 0.001.

**Figure 4 genes-13-02316-f004:**
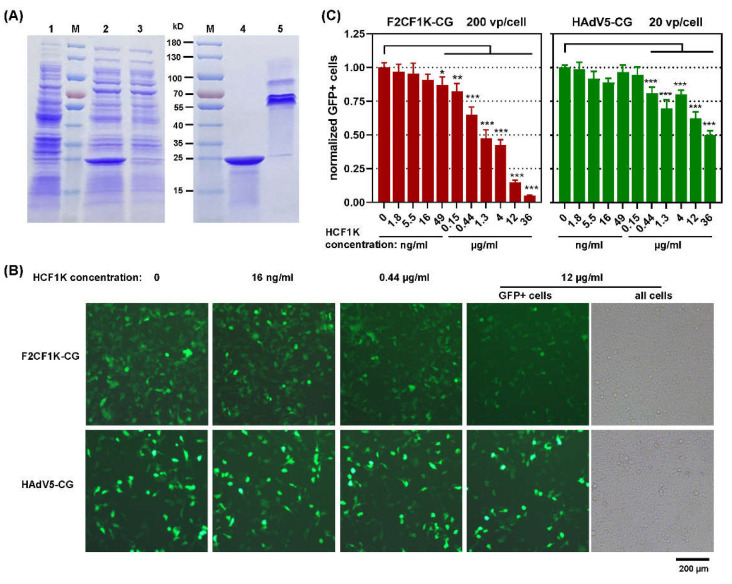
Block the infection of F2CF1K-CG or HAdV5-CG to 293 cells with recombinant CELO fiber1 knob. (**A**) Coding sequence of CELO fiber1 knob was inserted into pET-30a(+) prokaryotic expression vector. HCF1K (6× His tagged fiber knob) in soluble form was harvested from IPTG-induced Escherichia coli BL21(DE3) cells and purified with the method of immobilized metal ion affinity chromatography (IMAC). Samples collected from different stages of purification process were mixed with 2× SDS loading buffer, resolved by SDS-PAGE and visualized by staining. The loaded samples were from: 1. bacteria cells collected before addition of the inducer IPTG; 2. soluble total proteins from IPTG-induced bacteria; 3. flow-through of IMAC; 4. purified protein (boiled) and 5. purified protein (unboiled). (**B**) Human 293 cells were incubated with recombinant fiber knob at indicated concentrations for 30 min before infected with F2CF1K-CG at an MOI of 200 vp per cell or HAdV5-CG virus at an MOI of 20 vp per cell for 1 h. GFP+ cells were photographed under a fluorescence microscope at 24 h post-infection. (**C**) The intensity of GFP expression in each well was further measured on a multifunctional microplate reader and normalized to that in wells without the addition of HCF1K. All of the experiments were performed in duplicate and the data shown are from one representative experiment out of the three performed. * *p* < 0.05, ** *p* < 0.01 and *** *p* < 0.001.

**Figure 5 genes-13-02316-f005:**
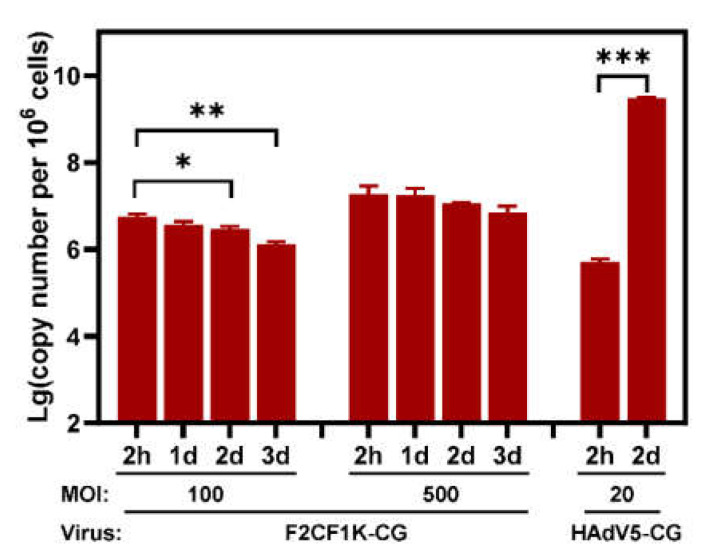
Detection of viral genome in adenovirus infected 293 cells. Human 293 cells in 6-well plates were infected in duplicate with F2CF1K-CG at MOIs of 100 or 500 vp/cell for 2 h. Total DNA in cells was extracted immediately after incubation (2 h) or at indicated time points (1, 2 or 3 days) post-infection. Taqman probe-based real-time PCR was performed in triplicate to determine the copy number of viral genome by detecting the GFP reporter gene. The cell number was similarly calculated after determining the copy number of human RNase P gene by real-time PCR. The copy numbers of viral genome in 10^6^ cells were calculated and shown. Human adenovirus HAdV5-CG served as a positive control. One-way analysis of variance (ANOVA) was applied for data from each F2CF1K-CG group of different MOIs after logarithmic transformation, with unpaired *t*-test for that from HAdV5-CG group. * *p* < 0.05, ** *p* < 0.01, *** *p* < 0.001.

**Figure 6 genes-13-02316-f006:**
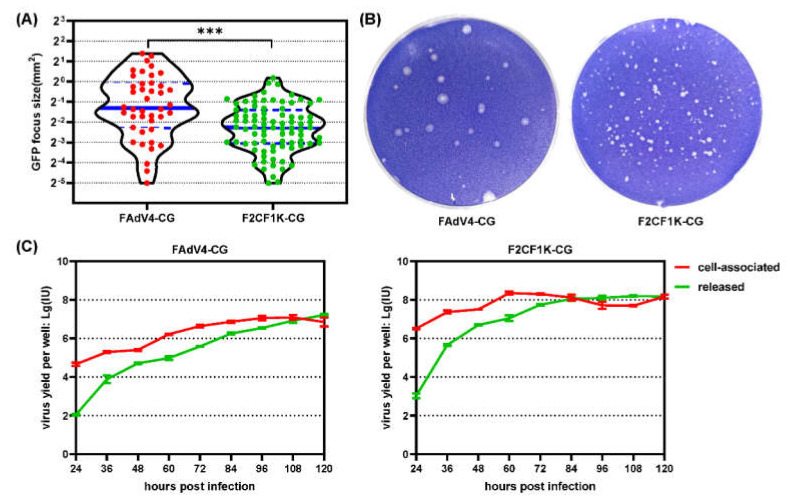
Growth of fiber-pseudotyped FAdV-4 vector in packaging LMH cells. Plaque-forming experiments were performed on chicken LMH cells. LMH cells in 6-well plates were infected with F2CF1K-CG or control FAdV4-CG at 10,000 vp/well for 2 h and maintained in semi-solid culture media. The GFP foci were photographed under a fluorescence microscope 5 days post-infection, and the areas of GFP foci were measured using the Fiji image processing package. Shown are the violin plots of the GFP focus size data. The size medians were compared by using non-parameter Mann-Whitney U test (**A**). The cells were fixed in paraformaldehyde and the plaques formed were visualized by crystal violet staining 6 days post-infection (**B**). The replication of F2CF1K-CG was further investigated by drawing one-step growth curve. LMH cells in 12-well plates were infected with F2CF1K-CG or FAdV4-CG at an MOI of 5 vp/cell for 2 h. Culture media and infected cells were harvested at indicated time points. The yields of progeny viruses associated with cells or released to the culture media were quantified by titration (**C**). *** *p* < 0.001.

**Figure 7 genes-13-02316-f007:**
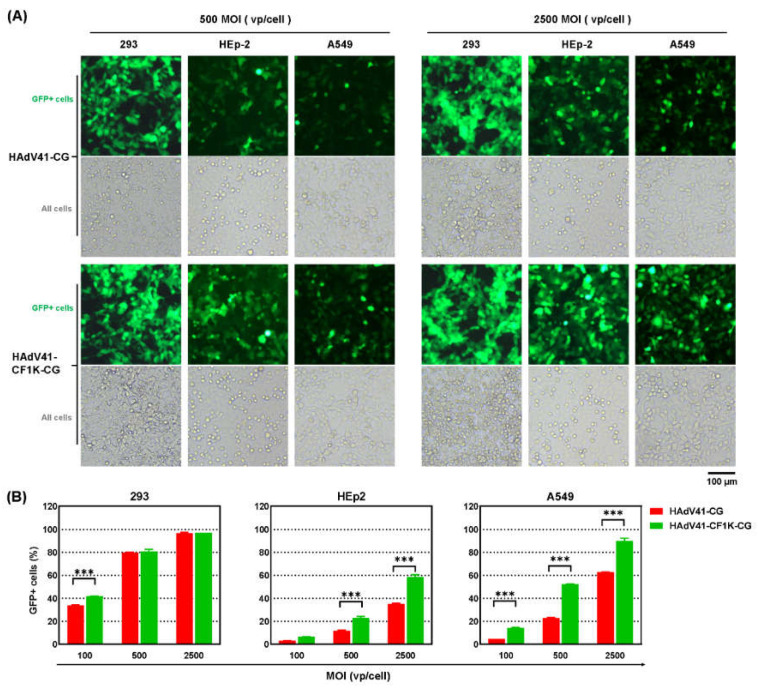
Transduction of human adherent cell lines with fiber-modified HAdV-41 vector carrying GFP reporter gene. (**A**) Human 293, HEp-2 and A549 cells were infected with HAdV41-CF1K-CG at an MOI of 100, 500 or 2500 vp/cell for 4 h, and the expression of GFP was observed under a fluorescence microscope at 48 h post-infection. HAdV41-CG, a E1/E3-deleted recombinant HAdV-41 carrying CMV promoter controlled GFP expression cassette, served as a control. HAdV41-CF1K-CG was different from HAdV41-CG in the CELO fiber1 knob (CF1K)-pseudotyped short fiber. (**B**) The percentages of GFP+ cells were further determined by flow cytometry. All of the experiments were performed in duplicate and the data shown are from one representative experiment out of the three performed. *** *p* < 0.001.

**Table 1 genes-13-02316-t001:** Summary information of PCR primers.

Fragment	Primername	Sequence	Template	Product (bp)
F2-AvrII	2206FAV4F2-CF1K1	cttacggtct ccgccaatgg ccttgggctg aagtacgaca ct	pMD-FAV4FS	103
	2206FAV4F2-CF1K2	gtgtggaact tcccccccct ccgaccacgg tta		
CF1K	2206FAV4F2-CF1K3	gagggggggg aagttccaca cccgaggtg	CELO genomic DNA	587
	2206FAV4F2-CF1K4	gggatcgaag aagttagtac ccgaggagtt c		
T-KpnI	2206FAV4F2-CF1K5	gaactcctcg ggtactaact tcttcgatcc c	CELO genomic DNA	114
	2206FAV4F2-CF1K6	ggacagctgt agagtcattg atagtacccc agataagtaa acg		
F2-HindIII	2206FAV4F2-CF1K7	ggtactatca atgactctac agctgtccag cggcct	pMD-FAV4FS	153
	2206FAV4F2-CF1K8	gattggacgc gggaacaaag gagag		
EG-SwaI	2207F2CF1KEGf	cgtcctttcg ttacagatct tcct	pKFAV4F1IJR-EG	2126
	2207F2CF1KEGr	cggtggatcg gatatcttat ctaga		
GFP-MluI	2208MluI-GFP1	gtcagatccg ctagagatct gctacgcgtg ccaccatggt gagcaagg	pKFAV4S-F2CF1K-CG	790
	2208MluI-GFP2	tctagatccg gtggatcgga tacgcgttag agtccggact tgtacagctc		
Ad41f-Kan	2208Ad41f-kan1	ctaccacaga aatgtccata ttatttaaat aaaacatcag gggctgagg	pKAd41CMV-MluI	
	2208Ad41f-kan2	aaataaaaca tcaggggctg aggtttaaac gtatactggc ttaactatg		
	2208Ad41f-kan3	actttaatta aaggggagaa gttccctgag gtttaaacgc gcgcaaactg		2695
	2208Ad41f-kan4	ttaaggtaag ctttattaat cagataactt taattaaagg ggagaagttc		2747
SF-PacI	2208Ad41SF-CF1K1	ctcagggaac ttctcccctt taat	pKAd41CMV-MluI	919
	2208Ad41SF-CF1K2	aggtggccac agtgagggat ggtacacttc gag		
S-CF1K	2208Ad41SF-CF1K3	atccctcact gtggccacct atcactgcg	CELO genomic DNA	655
	2208Ad41SF-CF1K4	gtagcaaaat acagctcatt gatagtaccc cagataagta aac		
SF-BglII	2208Ad41SF-CF1K5	gtactatcaa tgagctgtat tttgctacat aactgaacaa t	pKAd41CMV-MluI	333
	2208Ad41SF-CF1K6	cttccacgct agcatctgaa gaaag		
GFP-frag	2008GFPf	gacaaccact acctgagcac cc	Virus genomic DNA	126
	2008GFPr	cttgtacagc tcgtccatgc c		
	2008GFPprobe	HEX-tccgccctga gcaaagaccc caac-BHQ1		

**Table 2 genes-13-02316-t002:** Summary information of purified FAdV-4 and HAdV-41 vectors.

Virus Name	Fiber Modification	Promoter of Transgene	Physical Titer (×10^11^ vp/mL)	Infectivity Titer (×10^9^ IU/mL)	Particle-to-IU Ratio
FAdV4-CG	None	CMV promoter	35	8.4	420
F1IJR-CG	Insert RGD4C peptide in FAdV-4 fiber1 IJ loop	CMV promoter	6.6	1.3	520
F2CF1K-CG	Replace FAdV-4 fiber2 knob with CELO fiber1 knob	CMV promoter	10	25	40
F2CF1K-EG	Replace FAdV-4 fiber2 knob with CELO fiber1 knob	Human EF1a promoter	19	39	48
HAdV41-CG	None	CMV promoter	0.52	0.30	180
HAdV41-CF1K-CG	Replace HAdV41 short fiber knob with CELO fiber1 knob	CMV promoter	1.8	1.2	140

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
