# Peer review of "CELO Fiber1 Knob Is a Promising Candidate to Modify the Tropism of Adenoviral Vectors"

_genes, 2022, doi:10.3390/genes13122316_

Round 1
Reviewer 1 Report
Y Sun, et al. propose to use the avian CELO fiber1 knob to modify the tropism of human adenoviral vectors, specifically human adenovirus (HAdV) type 41. This will allow their use in humans for various gene transfer/vaccine purposes. On the surface, this would seem unlikely, as adenoviruses from birds and humans represent two different genera: Mastadenovirus and Aviadenovirus. There haven't been any reports of cross-genera infections in the “wild”, as global surveillance for this is very intense and long-standing.
I am not convinced the cross-genera chicken fiber knob inserted into FAdV-4 allows for recognition of the human cell receptor for human adenoviruses as reported here, given the data presented. In part, since their vector is HAdV-41. This human adenovirus contains two fibers and the researchers note replacing the short fiber only. Curiously, in one of their expts, HAdV-5 is used “as a control”. This is puzzling. Why as a control? Traditionally, HAdV-5 is used as a human gene transfer vector. If the authors were to go this route- Why wasn’t the “one fiber knob containing” HAdV-5 used as the vector of choice? Alternatively- Why wasn’t the second (long) fiber of HAdV-41 deleted in their attempt? With two fiber knobs in HAdV-41- The question is whether the recombinant bird fiber knob is the actual human cell entry “key” or is it the remaining human long fiber knob? I don’t remember specifically, but I was under the impression the long fiber knob was the identified host cell receptor binding version. (Though this may explain their paragraph "3.3. CELO fiber1 knob might have a cellular receptor different from CAR in human cells”).
While avian AdVs would certainly solve the long-standing problem of pre-existing immunity to human adenoviruses (HAdVs), it is not clear in this manuscript that the proposed avian fiber knobs interact successfully with human cell entry receptors.
For background, their initial trials with FAdV-4, presumably a bird AdV, failed so they reasoned that an exchange with another bird fiber knob, from Chick embryo lethal orphan virus (CELO), may work better. This is in contrast to the two citations, from two independent research groups, that note "….human cells although following studies doubted that [21,22]" vs one citation, from a third group using a mutant version to demonstrate "CELO has been constructed as a gene transfer vector and it could transduce human cells [14]".
Figure 2 shows "Adenoviral plasmids were linearized and transfected to chicken LMH cells, with GFP expression noted qualitatively. This is fine showing their construct works, in avian cells. (the listing of MW of bands is not necessary in the text). What about the equivalent in human cells?
The key to this paper is Fig 3- “Transduction of human adherent cell lines with fiber-modified FAdV-4 vectors carrying GFP reporter gene”.
Fig 3 needs a more informative legend- what are the F1IJR-CG, F2CF1K-CG, and F2CF1K-EG contracts? What is HAdV5-CG? Purpose in “some tests”? Why is it not shown in panel A? If HAdV-41 is the vector backbone, where is it in panels A and B, as a control? (Again, this would speak to if the second fiber is the actual host cell receptor binding protein).
If "F1IJR-CG, a previously constructed FAdV-4 carrying RGD4C-modified fiber1 served as a control"- A control for what? Panel A shows ~ no difference between this control and the three "tests". Also, for the “10000” data points, is there no data for HAdV5-CG?
This section is curious: "3.3. CELO fiber1 knob might have a cellular receptor different from CAR in human cells"- Does this mean the cell entry experiments were unsuccessful and the researchers are rationalizing the data? If so, then the "new, proposed" cellular receptor counterpart should be the target of their vector development.
Fig 4. In Panel A, the gel shows a cloudy blur at the top- ??? Also why is HAdV-5 the control rather than HAdV-41, if 41 is the vector backbone?
Fig 7 shows expts on the transduction of three human cell lines with HAdV-41 constructs. What is the point of this? Shouldn't this include the avian knob constructs? Shouldn't these be in Fig 3 as opposed to HAdV5-CG? Again, the legend should describe HAdV41-CF1K-CG and HAdV41-CG. If HAdV41-CG is the control, why is HAdV41-CF1K-CG “better” at transducing (higher GFP+cells %)? (It probably doesn’t matter, but most similar expts I’ve seen are in triplicate rather than duplicate).
An additional thought-
The text/Table 2 notes "HAdV-41 vectors- replace HAdV41 short fiber knob with CELO fiber1 knob"- is this the vector used for the constructs? If so, shouldn't this be one of the controls? Table 2 needs a more informative table legend, moved to the top of the Table.
Reviewer 2 Report
In this submission, Sun et al describe the development of a novel adenovirus system for transduction of human cells with a reduced immune recognition and the propogation of this virus system in LMH cells. The authors use fibre 1 in Fowl AdV-4 and CELO fibre1 knob and show improved infection in several cell lines. They also move the short knob to HAdV41, for which there is established genetic systems, and show significant infection of human cells. This submission shows the potential for CF1K use in human adenovirus transduction systems and may improve in vivo infections due to a lower likelihood of immune recognition.
The paper itself is well written and organized, figures are clear and strong and the authors conclusions are well supported.
References or evidence that there is no cross reactivity which may produce immunity with FAvD-4 would further strengthen the authors hypothesis.
Reviewer 3 Report
In this manuscript, Sun et al report that the CELO fiber1 knob domain can increase the transduction efficiency of non-human (FAdV-4) and human (HAdV-41) adenovirus-based vectors in human epithelial cells. The manuscript has a satisfactory introduction with a good literature overview. However, several important changes or clarifications need to be introduced in the manuscript.
1. As shown in Figure 5. FAdV-4 genome could not replicate in 293 cells, while HAdV-5 genome could replicate in 293 cells. Why did authors choose 293 cells in Figure 3B and Figure 4 to compare transduction efficiency with FAdV-4 and HAdV-5 when one vector can replicate while the other cannot in this cell line? It is possible that there is already CPE of some cells after infection with HAdV-5 (GFP expression was determined 24 and 48 h p.i.; MOI 20/100/1000 vp/cell).
1.a. Having that in mind, Figure 4C requires an additional assay to demonstrate the potential of FAdV-4 binding to CAR (CAR blocking antibody or HAdV-5 fiber/knob inhibition before transduction with FAdV-4 or similar approach) in cell line different than 293 (in which both vectors do not replicate).
2. Figure 2A, Figure 3A, Figure 4B, Figure 7A – it would be useful to show bright field images. In this presentation of the results, it is possible that the lower fluorescence is a consequence of less confluency of the cells, and not a weaker infection with the virus.
3. Figure 3A; Figure 7A – some of the images are out of focus, the brightness/contrast between different fields of view does not appear to be equal. My recommendation is to improve the quality of the images.
4. Figure 5. Information about the statistical test is missing. How are samples normalized? Is the number of cells the same between different samples? Number of tehnical replicas?
Minor:
1. Correct: Figure 4A – sample names 1,2,3,5,6 in 1,2,3,4,5. Figure 6B – FAdV-CG in FAdV-4. In Table 2, is FAdV4-GFP same as FAdV-4CG?
2. Why was HAdV-41 chosen as a human adenovirus model? Namely, reference 10 says that HAdV-5 and HAdV-41 have comparable pre-existing immunity, which is high. In addition, HAdV-41 lacks RGD, which decrease transduction efficiency compared to other HAdV serotypes. An explanation is needed in the manuscript.
Round 2
Reviewer 3 Report
The authors have significantly improved the article. The conclusions related to the receptor have now been rearranged so that they are more hypotheses for the following investigations, which is the correct interpretation of the results obtained in this paper.
The answers to my questions are satisfactory. However, the design of two experiments (CELO fiber1 knob-pseudotyped FAdV-4 transduction of human adherent cell lines and CELO fiber1 inhibition of transduction) was certainly not good considering that the authors were using cell line in which one vector could replicate and the other could not. I emphasize that this kind of experiment design is not correct and must be avoided in the future. Yet, understanding that the authors made experiments that required time, consumption of materials and someone's hard work, and provided valuable data, I suggest that the authors keep the existing results in the form in which they were presented. But, since the autors agree with me (based on the response to revision; 1, 1a) and I believe that it is also in their interest to point out the following in a short comment in the manuscript:
A) Reasons why you are using 293 cells in Figure 4 (since 293 was the most sensitive cell line to the transduction of fiber-pseudotyped FAdV-4)
B) In the observed time 24 h p.i. at a low dose of 20 vp/cell HAdV-5 vector did not lyse 293 cells.
For me, the good place to explain that is in the line 368 after the sentence: 293 cells were incubated with HCF1K at serial concentrations and infected with F2CF1K-CG or HAdV5-CG.
Other possibility is to remove 293 cell line from Figure 3, and put there HEp-2 cells (from supplement). If so, the results and discussion text should be corrected accordingly.
Authors should choose one of the suggested possibilities.
